behaviour/robotics

swarm robotics, ants, stigmergy, pheromone, diffusive processes, scalability

**Authors for correspondence:**
Edmund R. Hunt
e-mail: edmund.hunt@bristol.ac.uk
Sabine Hauert
e-mail: sabine.hauert@bristol.ac.uk

# Testing the limits of pheromone stigmergy in high-density robot swarms

Edmund R. Hunt[1,2], Simon Jones[2] and Sabine Hauert[1,2]

[1]Department of Engineering Mathematics, University of Bristol, Merchant Venturers Building, Bristol BS8 1UB, UK
[2]Bristol Robotics Laboratory, University of the West of England, Frenchay Campus, Coldharbour Lane, Stoke Gifford, Bristol BS16 1QY, UK

 ERH, 0000-0002-9647-124X; SH, 0000-0003-0341-7306

Area coverage and collective exploration are key challenges for swarm robotics. Previous research in this field has drawn inspiration from ant colonies, with real, or more commonly virtual, pheromones deposited into a shared environment to coordinate behaviour through stigmergy. Repellent pheromones can facilitate rapid dispersal of robotic agents, yet this has been demonstrated only for relatively small swarm sizes ($N < 30$). Here, we report findings from swarms of real robots (Kilobots) an order of magnitude larger ($N > 300$) and from realistic simulation experiments up to $N = 400$. We identify limitations to stigmergy in a spatially constrained, high-density environment— a free but bounded two-dimensional workspace—using repellent binary pheromone. At larger $N$ and higher densities, a simple stigmergic avoidance algorithm becomes first no better, then inferior to, the area coverage of non-interacting random walkers. Thus, the assumption of robustness and scalability for such approaches may need to be re-examined when they are working at a high density caused by ever-increasing swarm sizes. Instead, subcellular biology, and diffusive processes, may prove a better source of inspiration at large $N$ in high agent density environments.

## 1. Introduction

Exploration and area coverage are classic challenges for cooperating robot teams: agents need to coordinate effectively such that an unknown territory is rapidly reconnoitred and then monitored on an ongoing basis. Applications include search and rescue, the deployment of communication networks, environmental monitoring [1] and even cancer treatment using nanobots [2]. In the swarm robotics paradigm, inspiration is taken from collective animal behaviour, for example ant colonies that are organized according to simple interaction rules based on local communication and information [1]. One form of communication,

stigmergy, allows robots to communicate through the environment, for example through depositing chemical markers known as pheromones that facilitate indirect communication and coordination [3]. Yet, just as in nature where stigmergy has limits to its usefulness, for example where army ants form futile circular mills through trail following [4], so too in robotics limitations may arise when the environment becomes saturated with pheromone information.

The focus of much biological research has been on the use of *attractive* pheromones that, upon detection, recruit group members to a valuable resource such as food [5], or to an intruder [6]. There has also been a more limited amount of research on *repellent* ('anti', negative or territory-marking) pheromones, which may be used by ants to mark an unprofitable foraging pathway [7], an unsuitable nest site [8] or an otherwise empty area around a colony that is nevertheless liable to be defended against intruders [9]. Such chemical markers may be deposited passively from ants' footprints, which leave behind a residue from the outer cuticle of the animal, which is coated in a signature mixture of hydrocarbons [10]. The research of one of the present authors (Hunt *et al.* [11]) indicates that such passive markers can expedite an ant colony's collective exploration of an unfamiliar territory. In the context of swarm robotics, deflective movement away from 'always-on' trail markers is a very simple rule that can be used for stigmergic coordination in a collective exploration and/or area coverage task. By moving away from 'pheromone' markers left by the swarm, significantly less time should be spent revisiting previously explored areas.

However, avoidance behaviour might be expected to reach the limits of its usefulness in larger groups of agents if they reach a high density. This is because agents may be unduly constrained by the large amounts of repellent pheromone being deposited by fellow group members. Swarm robotics approaches are often assumed to be robust and scalable, yet neither of these characteristics may be true when increasing $N$ to large numbers and a high robot density in a scenario where robots interact via stigmergy. This is in addition to the challenge of more robot–robot collisions, which beyond an optimal swarm density would be expected to reduce performance owing to physical interference [12–16]. Instead, high-density swarm performance may be limited by reliance on stigmergy, whereby it may become preferable for individuals to be non-interacting and ignore social information that has become uninformative. Thus, we anticipate and investigate the limits of stigmergic area coverage at higher swarm densities, relative to a non-interacting, random walk behaviour in the agents. We do this in simulation and with real robots, using the Kilobot platform. Section 2 provides a brief overview of previous biological and swarm robotics work on repellent pheromones, §3 describes the simulation and robotics methods, §4 presents results and §5 discusses the significance of our findings for the scalability of stigmergy-based control algorithms at large $N$ and high agent density.

## 2. Background and previous research

One of the principle advantages of the swarm robotics approach is that agents can have simple computational hardware, and hence be cheap and replaceable [17]. Stigmergy as a control mechanism greatly facilitates this goal, because sophisticated memory encoding, storage and retrieval can be substituted for chemical deposition and detection. Information transfer is possible via the modification of the environment. Indeed, stigmergy is a likely precursor to the development of internal memory [18], and has been observed as such in very simple organisms, including slime moulds [19]. Through this simple organizing principle, swarms of organisms or robots can exhibit remarkable, emergent 'swarm intelligence', with respect to tasks such as foraging or exploration [20]. Pheromones, chemical markers deposited in the environment, are a particular form of stigmergy and are used most notably by the ants. For example, a 'no entry' mark can be used to deter other ants from travelling down an unprofitable foraging route [7]. In animal communication, one distinguishes between a *signal*, sent or left deliberately in the environment to modify the behaviours of others, and a *cue*, which may also alter the actions of others, but which may be transmitted inadvertently, and at no cost to the sender. Pheromones are signals, but passively deposited chemical cues may also be of great importance to various species. For example, bees may leave behind scent marks in recently visited flowers from their footprints and use these to forage more efficiently [21]. Ants can use footprint cues to discriminate between the foraging trails of different colonies [22] and to avoid competing conspecifics or potential predators [10]. Footprint cues may also be used to avoid nest-mates when scouting for food sources [9], avoiding the redundant revisiting of previously explored space. Our previous research also indicated that ants may avoid the footprints of other nest-mates to collectively explore unfamiliar territory, even when not scouting for food (Hunt *et al.* [11]). An alternative form of stigmergy to

deploying chemicals is depositing building material, as seen in the termites; using this concept to coordinate a group of robots was previously found to show worsening task performance at larger group sizes (four or five versus three) owing to time-consuming robot interactions [23].

It is in the exploration of unfamiliar space, and in ongoing area coverage, that swarm robotics has most often drawn upon the concept of repellent markers. This has been tested through three main methods: digital pheromones deposited on a shared virtual map, simulation studies and real-world environments, generally using light patterns. We review some of this research as an overview of approaches. The method of implementation may also have some relevance to observed results, with, for example, real-world pheromone implementations being noisier to detect than digital ones.

## 2.1. Digital pheromones

Stigmergy-based coordination through pheromones can be achieved through deposition on a virtual map, with the map shared globally among the swarm members [24]. Digital deposition can also be made on the robots themselves, which record levels of attractive and repulsive pheromones being added to them from robots transmitting nearby. This has been demonstrated in 20 robots [25,26]. Such digital pheromones have been used to coordinate unmanned aerial vehicles (UAVs), with their use in 100 simulated UAVs reported in a US military war-game [27], and up to 20 simulated UAVs in another study [28]. Another study used a node-based representation of the spatial environment upon which robots deposited both attractive and repellent pheromone; this used eight simulated and two to four real robots [29]. A modified Kilobot system 'Kilogrid' has used a system of floor-based communication modules to demonstrate a pheromone-based foraging task in 50 real robots [30]. Similar capabilities have been developed in the 'augmented reality for Kilobots' (ARK) system by using an overhead controller to track robots in a virtual environment [31]. A recent study using this system considered virtual pheromone recruitment to four foraging patches with 100 real robots and up to 200 simulated robots [32].

## 2.2. Simulated pheromones

The third class of study has investigated pheromone swarm robotics in simulation and thus remains agnostic as to how such stigmergy may be implemented in practice. One simulation study examined the effectiveness of repellent pheromone with 3–24 robots [33]. Another study with 10 agents employed individual-specific pheromone trails and examined how group behaviour changed with varying sensitivity of agent to own versus others' pheromones [34]. This was complemented by a further simulation investigation of various performance metrics for stimergy-based area coverage. With 10 agents, it examined the impact of varying parameters, such as pheromone diffusion and evaporation rate, and different movement strategies (Brownian versus Lévy flights) [35].

## 2.3. Environmental pheromones

Pheromones can also be deposited into the spatial environment itself. This has often been done using projections of light: an early example showed ant-inspired trail-following and avoidance behaviour in five robots [36], while a more recent study used 10 [37]. Another technique using light involves manipulating the surface upon which the robots move. Pheromone trail following with up to six robots (one leader and five followers) has been achieved with LCD TVs [38], while phosphorescent-glowing paint has been demonstrated for trail following in one robot [39]. Thus, environment-based pheromone markers have not, until now, been implemented in very large, spatially constrained swarms (e.g. greater than 100 robots). High swarm density may result in large amounts of pheromone, thus making it harder to avoid, especially for robots with limited capabilities such as the Kilobot. This challenge is in addition to the higher rate of robot–robot collisions. Our approach is explained in §3.2.

The previously described studies use a relatively low number of real robots (generally, 2–20 robots, though 50 in [30] and 100 in [32]). The robot densities are summarized in table 1. We use Kilobots to investigate the operation of a simple stigmergic interaction rule in very large $N$ swarms: up to 324 real robots and 400 simulated.

**Table 1.** Robot swarm density of some previous research, compared to maximum investigated here.

| reference | type | vehicle | number | area | point density/m$^{-2}$ |
| --- | --- | --- | --- | --- | --- |
| this study | real | ground | 324 | 6 m$^2$ | 54 |
| this study | simulation | ground | 400 | 6 m$^2$ | 66.7 |
| [24] | simulation | air | 10 | 30 km$^2$ | 0.0000033 |
| [28] | simulation | air | 20 | ~1 km$^2$ | ~0.0002 |
| [36] | real | ground | 5 | ~0.25 m$^2$ | ~20 |
| [37] | real | ground | 10 | ~0.4 m$^2$ | ~12.5 |
| [38] | real | ground | 6 | 0.48 m$^2$ | 12.5 |
| [30] | real | ground | 50 | 2 m$^2$ | 25 |
| [32] | real | ground | 100 | 4 m$^2$ | 25 |
| [32] | simulated | ground | 200 | 6.25 m$^2$ | 32 |

# 3. Methods

## 3.1. Robot arena and pheromone

We use a $3 \times 2$ m arena for our experiments: a smooth surface made of white acrylic plastic, bordered with 5 cm walls. Kilobots are placed in the centre of the arena in a square grid formation (spaced 4 cm apart from their centre, or with a diameter of 33 mm, a gap of 7 mm), and they all execute the same simple controller (a homogeneous swarm) for 20 min. The initial formation of the robots in the centre of the arena can be thought of as the release of a robot swarm by a user into a territory for reconnaissance, or the injection of nanoparticles into a target site. The initial heading of the robots was randomized in the simulator (§3.3) and as far as possible in reality. Kilobots are simple, small and cheap robots that can thus be combined into large swarms [40]. Above the arena is a webcam (Creative VF0700 Live! Cam Chat HD), which has full view of the arena and can detect Kilobot positions using a simple image processing routine.

The identification of Kilobot positions and the projection of pheromone depositions took place in Matlab 2017b (using its image processing toolbox) and consists of a cropping step (removing area outside arena), a thresholding step to pick out the metallic Kilobots on the white arena and a circle detection step (Kilobots are 33 mm in diameter). There is also a projector above the arena, which creates a dynamic pheromone environment using light patterns. The Kilobot detection routine takes place twice per second, and circles of pheromone (blue light) are projected onto the detected locations with a 0.5 s delay, thus typically placing the pheromone slightly behind the robot as it moves. In the case that the Kilobots are executing a turning motion and the deposition occurs on top of the robot, this does not interrupt its behaviour. This is because, as noted in §3.2, previously chosen movements are completed before further sensing of the environment. This implementation of the closed-loop tracking and deposition process has the advantage of simplicity, exploiting the focus on swarm-level behaviour rather than individual-level tracking.

To sense whether the Kilobot is in a region of previously deposited pheromone, it needs to detect the colour of the local ambient light. This is sensed using the upwards-facing phototransistor of the Kilobot, according to the pulse length characteristics of light from low-cost digital light processing (DLP) projectors [41]. This allows four distinctive colour groupings to be sensed (primary colours, cyan and yellow, magenta and black). The method used is described in [42]. We use blue light to 'deposit' pheromone behind the moving robots, which can then be detected and responded to by both the depositing individual and its fellow 'nest-mates'. The detection of light was binary (detected or not) rather than on a continuous scale. Thus, as the simulated pheromone 'evaporated' over time, it remained projected with full brightness until it reached a 0.5 threshold on a 0–1 scale, when it was turned off. This was implemented to give a decay time of 140 s (see also §3.3). This decay rate was chosen as an initial starting point for the investigation, as it appeared to allow pheromones sufficient time to affect swarm behaviour for medium-sized groups ($N = 64$), in relation to the (relatively slow) movement speed of Kilobots. Pheromone was deposited continuously by all robots, regardless of

whether they were already in regions where pheromone had been laid. In this case, the pheromone concentration would return to the maximum concentration (i.e. a full 140 s seconds of decay time). Robots could not tell the difference between their own and others' pheromone.

## 3.2. Robot controller

The Kilobots were programmed with a simple finite state machine: either engage in random walk-type movement behaviour, or if pheromone is detected, engage in simple scattering (avoidance) behaviour. Because the Kilobot has limited sensory and motor capabilities [40], it is not possible to strictly 'avoid' pheromone marker when it is detected, in the sense that a path away from it cannot be planned out and navigated precisely. Essentially, the Kilobot may perform a repertoire of random walks [43]. Nevertheless, a change in random walk behaviour was found often to be effective in deflecting it from a pheromone patch when encountered. The random motion in both the random walk and scattering behaviours also helps the Kilobots move away from the arena walls if they are in that position, which they are not otherwise aware of as a result of their purposefully simple design.

The standard Kilobot controller operates via an execution loop that can repeat every specified number of internal clock ticks: each tick is approximately 30 ms, and so we set the loop timer to 16 ticks or just shy of 0.5 s. Initially, implementation of the sensing function obtained a reading from the environment every loop iteration, but this was found to result in ineffective Kilobot exploration performance, because frequent switching between behaviours resulted in trapping in circular movement patterns or at the arena barriers. Sensing of the environment at the end of a previous movement, before undertaking a relevant movement (scattering or random walk) for a predetermined length of time, was found to be much more effective, especially in our experimental context, when rather large amounts of pheromone could be present. This has interesting parallels with previous research on motor planning in animals, which noted various potential benefits from intermittent movement and sensing of the environment to reduce information-processing demands [44,45].

For the random walk behaviour, a random number is generated in the set {0,1,2}. If the number is 2, the Kilobot's two motors are both set to on, which results in a (roughly) forwards motion; otherwise, it turns to the left on 0 or right on 1. Thus, forward motion occurs around 33% of the time on average and left and right turning 33% each. Each period of movement lasts for 1, 2 or 3 s with equal probability.

The scattering (avoidance) behaviour is coded as follows. If pheromone (blue light) is detected, the Kilobot randomly turns either left or right for 0.5 s and then it goes forward for 0.5, 1 or 1.5 s with equal probability. This often has the effect of deflecting the robot away from the detected pheromone trail (figure 2) and can be compared to the run-and-tumble movement dynamics in bacteria such as *Escherichia coli* [46,47]. However, as noted later in the results, this scattering behaviour actually results in a relative decrease in area coverage performance at higher swarm density (larger $N$). If pheromone is still detected, this procedure repeats; otherwise, the Kilobot reverts back to the random walk behaviour. For the control group of experiments, the same controller was used, but the environmental sense step was fixed to detect no pheromone; as a result only the random walk behaviour took place. We simulated the Kilobot behaviour for various numbers of robots and collected data on real robots for $N = 64$ (an $8 \times 8$ grid layout) and $N = 324$ ($18 \times 18$). There were five real robot trials for each experimental treatment and group size, for a total of 20 trials.

## 3.3. Simulations

### 3.3.1. Baseline scenario

We investigated the collective performance of the stigmergic area coverage algorithm with different numbers $N$ of robots. Because the Kilobots were arranged in a grid formation, we simulated square numbers of robots: $N = 1, 4, 16, 64, 144, 256, 324$ and $400$ (up to $20 \times 20$ robots). These group sizes resulted in swarm densities as shown in table 2. Point density is the number of robot entities per m$^2$ in the 6 m$^2$ arena, while physical density is the proportional occupation of the arena by robot hardware (each robot being 33 mm in diameter or $8.5 \times 10^{-4}$ m$^2$ in area). Simulations were carried out in an in-house simulator 'Kilobox' [42,48], based on the physics simulation library Box2D [49], which permits accurate simulation of Kilobot behaviour based directly on their real controller code. The simulated trials were carried out $2 \times 30$ times for each group size, first using the controller described previously and then as a control group without the avoidance behaviour (i.e. random walk only). Pheromone deposition was controlled via four parameters: these determined the rate of deposition (this was set to be maximal, i.e.

**Table 2.** Density of robots at the different N tested.

| N (robots) | density in 3 × 2 m = 6 m² arena | |
| | point (robots m⁻²) | physical (2D area occupation) |
| --- | --- | --- |
| 1 | 0.167 | 0.00014 |
| 4 | 0.667 | 0.00057 |
| 16 | 2.67 | 0.0023 |
| 64 | 10.7 | 0.0091 |
| 144 | 24 | 0.021 |
| 256 | 42.7 | 0.037 |
| 324 | 54 | 0.046 |
| 400 | 66.7 | 0.057 |

initial saturation on a 0–1 scale), distance behind the robot of the deposition (slightly behind), the size of the circular pheromone patch (the same size as the robots) and the exponential decay constant. This final parameter was set to $r = 0.005$, and the simulated Kilobots made a binary detection of pheromone (i.e. not detected or detected) with a threshold of 0.5 on a 0–1 scale. Together with the decay constant, this resulted in a decay time to below 0.5 in around 140 s ($\ln(0.5/0.005) = 138.6$ s). Simulations were carried out on a Late 2016 MacBook Pro (2 GHz Intel i5, 8 GB RAM) and run times for the pheromone-based simulations ($T_{sim} = 20$ min) ranged from around 2 s per trial for $N < 25$, up to 4 min for $N = 400$ (or 2 h for 30).

### 3.3.2. Further investigations

In addition to the baseline scenario described in the previous subsection, where the pheromone had a decay constant $r = 0.005$, and robots explored a $3 \times 2 = 6$ m² arena and laid pheromone continuously, we made the following further investigations.

*Constant density, lower group size.* To further examine whether observed trends with increasing N are a result of increased density, or increased group size, we repeated the analysis in a smaller arena, of half the area of the original with the same aspect ratio, size $\frac{3}{\sqrt{2}} \times \frac{2}{\sqrt{2}} = 2.12 \times 1.41 = 3$ m² and half the numbers of robots, thus maintaining the increasing density trend but halving the group sizes.

*Pheromone deposition threshold.* In the baseline scenario, pheromone was laid continuously, and this seems likely to contribute to oversaturation of pheromone at high density. We tested the effect of only laying additional pheromone when robots measured it to be below a certain threshold level. We set this threshold to be 0.75, where 1 is full saturation (maximum pheromone in a location), which is the level at which it is laid. That is to say, fresh pheromone had to decay for a period of time before it could be overlaid again.

*Decay parameter sensitivity.* To perform a pheromone decay parameter sensitivity analysis for each of the scenarios (baseline, smaller arena and deposition threshold), we repeated the simulation experiments with the decay parameter set to $r = 0.010$ (decay time of 69.3 s) and $r = 0.0025$ (decay time of 277.2 s), i.e. both a faster and slower decay of pheromone than $r = 0.005$ (decay time of 138.6 s).

The full results of these further investigations are presented in the electronic supplementary material, while the results in the main text (§4) generally refer to the baseline scenario described in §3.3.1.

### 3.4. Data analysis

The simulations were output into text log files, recording the position of each Kilobot every 1 s. These were analysed using Matlab 2017b, using a simple script, to read the files and calculate the following metrics: (i) area coverage on an ongoing basis, and (ii) cumulative proportion of the arena explored. Tracking of the individual robots' identity was not necessary as performance was assessed at the group (swarm) level. The area coverage was measured as the number of distinct cells occupied in the foregoing 10 s window where the cells were 20 × 20 cm (400 cm²). This is because the communication range of the Kilobots is approximately 10 cm and thus covers an area roughly 20 cm

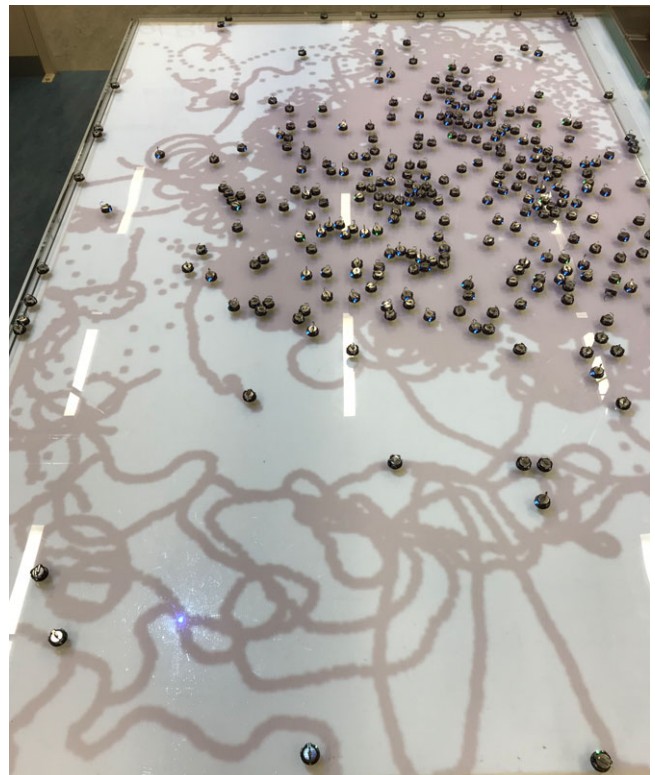

**Figure 1.** Arena at $T = 15$ min, $N = 324$ Kilobots in the trail avoidance (stigmergy) condition. A large patch of repellent pheromone has formed.

in diameter; this is taken as a proxy for a robot sensor range. The exploration of the arena was calculated as the percentage of arena visited at least once by any robot, given a division into $10 \times 10$ cm ($100$ cm$^2$) cells. This represents a static rather than a dynamic environment, where changing conditions require repeated reconnaissance. These two metrics are complementary yet capture quite distinct aspects of performance: although exploration may occur effectively, with most of the arena being passed through initially by the robots, area coverage may be poor in the longer term with crowding into corners, for example. On the other hand, area coverage might be quite good, with the robots spread out fairly widely, and yet with some parts of the environment never being reached. Performance metrics were calculated for the real Kilobot swarms by using coordinates recorded from the same process generating the stigmergic overlay from camera images. We fitted logistic curves to the change in area coverage and exploration over time to obtain estimates of the growth rate $k$ and the maximum value $L\_max$. These estimates were then used to obtain quantitative, statistical comparisons of the performance of the robots using pheromone coordination versus those without. These two logistic curve parameters $k$ and $L\_max$ capture complementary insights into swarm dynamics: initial speed of area coverage and exploration (shorter-term) and longer-term performance.

## 4. Results and discussion

Robot swarms coordinating through stigmergy outperform non-interacting robots at an area coverage and collective exploration task, but only within a certain swarm density range. A photo of the experimental arena for $N = 324$ Kilobots for an experimental trial at $T = 15$ min into the 20 min task, using stigmergy, is shown in figure 1. This illustrates at least one important observation about the performance of stigmergy at high swarm density. Performance is impaired because the pheromone avoidance behaviour begins to impede the passage of Kilobots past each other. This is because broad patches of pheromone are formed, as seen in the top half of the photo. This 'traps' a large number of agents within the same region. At the same time, performance is not completely impaired because a smaller number of robots on the periphery are able to encounter empty space that is not already saturated with pheromone information. In the bacterial world, domains of high density can similarly form where many bacteria engage in run-and-tumble dynamics in a crowded environment [46,47], a prelude to biofilm formation. Although

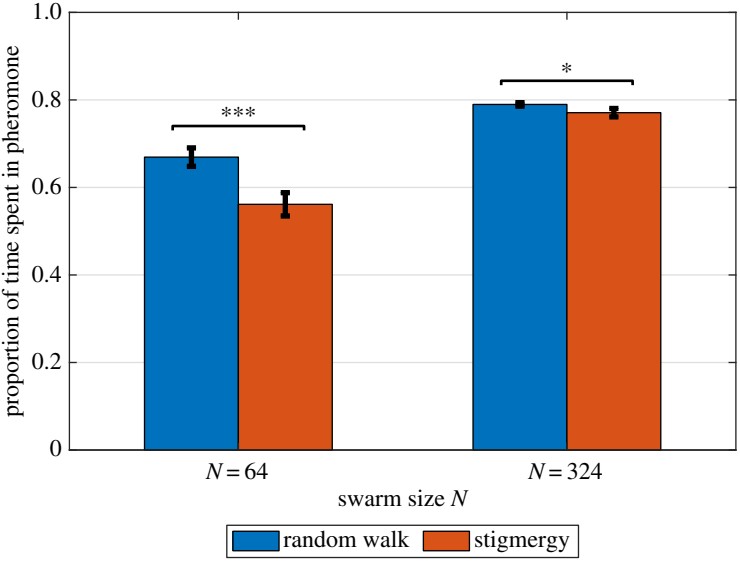

**Figure 2.** Mean proportion of time spent in pheromone by real robots, in control (random walk) and stigmergy treatments. At swarm size $N = 64$, robots in the stigmergy treatment spend significantly less time in the pheromone. At $N = 324$, there is a significant, but smaller difference. Error bars show standard deviation.

dense patches of pheromone are deleterious to area coverage and exploration performance, depending on the context, a high concentration of entities could be highly desirable (e.g. targeted nanoparticle drug delivery). Other research has found that obstacles can be used for controlling large populations of robots [50], and such patches could be seen as an emergent control mechanism.

Figure 2 indicates that stigmergy results in notably less time spent in pheromone in real robots at $N = 64$ (on average 56.1% of the time compared to 66.9%, a decrease of 16.1%; unequal variances $t$-test: $t = -6.34$, $p < 0.001$) but less so at $N = 324$ (on average 77.1% of the time compared to 79.0%, a decrease of 2.4%; $t = -3.62$, $p = 0.013$). Figure 3 and table 3 show that at $N = 64$, the swarm using stigmergy has an advantage at area coverage: spreading out to occupy different parts of the arena on an ongoing basis. The advantage at $N = 64$ of using stigmergy appears clear, with area coverage of around 36% compared to 30%, and exploration of 90% of the arena at $T = 20$ min, compared to 72% for the random walkers. At $N = 324$, there is no ultimate advantage to either area coverage or collective exploration for using stigmergy, with both methods reaching a similar area coverage of around 65%, and 97% exploration at the same time of $T = 1000$ s or just under 17 min. Indeed, for area coverage, sometimes the median coverage of random walkers exceeds that of the robots avoiding pheromone trails. Nevertheless, the pheromone avoidance method has some advantage in faster initial dispersal.

The robotic results were qualitatively confirmed in simulation (carried out at a range of group sizes, $N = 1-400$, see electronic supplementary material, figure S1). These indicated that at $N = 4$ onwards, there is a clear improvement in collective exploration (visiting each part of the arena at least once) when using stigmergy (figure 4; electronic supplementary material, figure S1).

This cumulative exploration advantage is maximized at $N = 144$. However, at $N = 144$, there appeared to be no gain in area coverage performance, and at even larger sizes of $N = 256, 324, 400$, stigmergy resulted in *inferior* performance to non-interacting random walkers (figure 4 and table 4; electronic supplementary material, S1). This corresponds to effective performance at a physical robot density up to around 2% of the physical space (table 1), while beyond this at 3%+, the stigmergy-based movement rule is a hindrance. It is worth noting that in simulation for $N = 324$, the longer-term performance ($L\_max$) is statistically better for the null model (random walk only) for both area coverage and exploration metrics (table 4); a significant difference in this parameter is not seen in the real robots (table 3). This appears to be because of more effective random walk dispersal in the simulator in the case of area coverage, and better pheromone dispersal in the real robots in the case of exploration. In both cases, this could arise from more noisy behaviour in the real robots than in the simulator, but the point about diminished performance (relative to non-stigmergy) at high swarm density is confirmed. We also observe that the cumulative exploration curves rise more steeply (higher growth rates $k$) in the real robots than the simulated ones: again, this could be because in large swarm sizes, a greater diversity of behaviour in the real robots means a few particularly exploratory

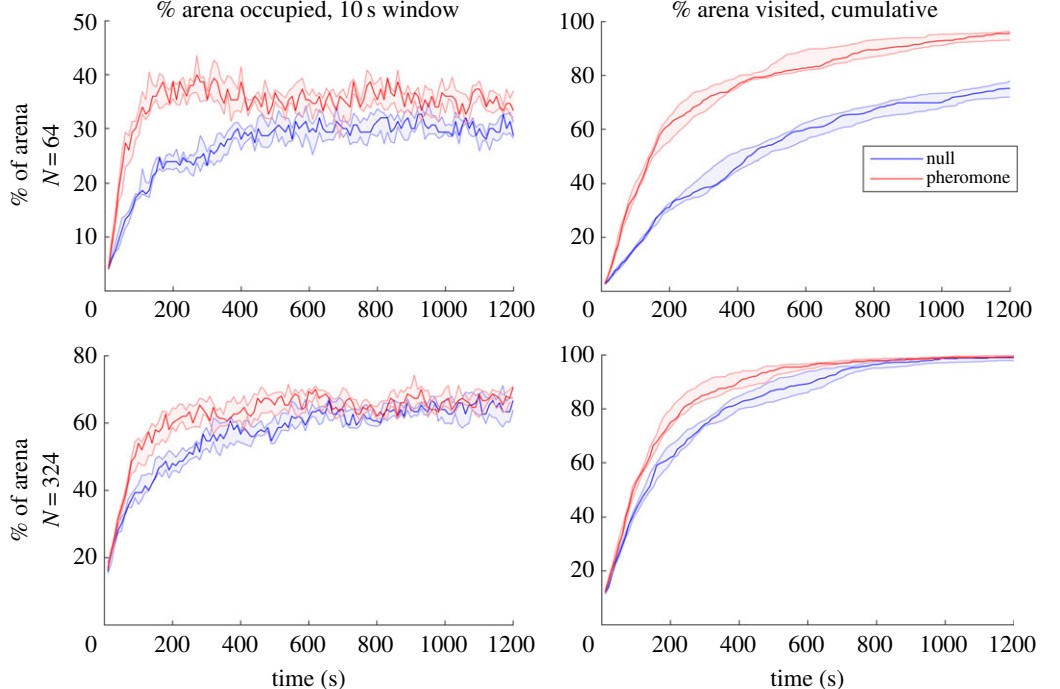

**Figure 3.** Performance of real robotic swarms ($N = 64$, 324). Median and lower and upper quartiles shown for five trials per condition. At $N = 64$, there is a significant improvement in area coverage and exploration using stigmergy. At $N = 324$, this advantage is much reduced; for area coverage, sometimes the median coverage of random walkers even exceeds that of the robots avoiding pheromone trails. Thus, diffusive processes may be more suitable for large robotic swarms in constrained environments, which may be compared to processes seen at the subcellular level.

individuals result in faster swarm-level exploration than anticipated. In this case, greater behavioural diversity associated with larger groups seems to be an asset.

A faster pheromone decay rate helps to reduce, though not eliminate, the negative effects on long-term area coverage and exploration at high swarm density (electronic supplementary material, figure S2 and table S2; cf. electronic supplementary material, figure S1 and table S1). Following the logic of decreasing pheromone longevity as swarms get denser, it presumably makes sense to turn pheromone off completely at sufficiently high density. A slower decay rate exacerbates the performance issues at higher density (electronic supplementary material, figure S3 and table S3).

The swarm dynamics in a smaller simulated arena, with the same swarm density (smaller group size), were broadly similar to the baseline scenario results (electronic supplementary material, figures S4–S6 and tables S4–S6). Indeed, the random walk (non-stigmergy) treatment had a stronger advantage over pheromone treatment at higher densities, which seems to point toward the relevance of diffusive processes in smaller scale systems.

In our implementation of pheromone laying and detection, the robots do not differentiate between their own and others' pheromone trails. In some cases, ants are thought to use individual-specific trail pheromones for orientation outside their nests [51] or to measure area inside nests [52]. If individual robots in our exploration and area coverage scenario were able to distinguish their own trails, it may help them to avoid getting 'stuck' in high-density pheromone regions. With individual-specific pheromones, it may also be possible to implement alternative, individual-specific avoidance strategies. However, individual-specific pheromones would require individual-level tracking and distinctive projector colours for each robot. With a colour palette of four detectable colour types (see §3.1) in our set-up, this would not be practicable, though perhaps a larger palette could be implemented using colours that pulse at characteristic frequencies (detectable by the robots but not by a human observer). However, this would limit the scenario's relevance to other contexts where individual trail laying and detection would not be possible, particularly at the micro or nanoscale, where one might deploy light-sensitive particles for example [53]. Individual trail identification may also limit the scalability of swarm systems. In the 'Kilobox' simulations, saturation gradients of pheromone (on a scale of 0–1) are present, but in the real robots this is implemented as a binary marked/absent light. It may be possible to sense light intensity gradients using the Kilobots' ambient light sensor, though this would

**Table 3.** Performance of real robotic swarms ($N = 64, 324$) in terms of fitted logistic curves with parameters $L\_max$ (maximum value) and logistic growth rate $k$. Standard error shown, and result of unequal variances $t$-test comparison between treatments.

| | $N = 64$ | | | | $N = 324$ | | | |
|---|---|---|---|---|---|---|---|---|
| | null | pher | $p$ (diff) | $t$ | null | pher | $p$ (diff) | $t$ |
| *area coverage* | | | | | | | | |
| $L\_max$ | $30.22 \pm 1.39$ | $35.64 \pm 0.55$ | <0.001 | −7.27 | $63.93 \pm 2.31$ | $65.55 \pm 1.84$ | 0.31 | −1.10 |
| rate $k$ | $0.011 \pm 0.004$ | $0.047 \pm 0.011$ | 0.002 | −6.31 | $0.008 \pm 0.002$ | $0.020 \pm 0.003$ | <0.001 | −6.78 |
| *exploration* | | | | | | | | |
| $L\_max$ | $72.08 \pm 4.87$ | $90.03 \pm 2.55$ | <0.001 | −6.53 | $96.23 \pm 1.83$ | $97.26 \pm 0.82$ | 0.35 | −1.03 |
| rate $k$ | $0.006 \pm 0.001$ | $0.009 \pm 0.002$ | 0.009 | −3.77 | $0.008 \pm 0.002$ | $0.013 \pm 0.002$ | 0.009 | −3.98 |

presumably be noisier than the binary detection event of full brightness pheromone versus no pheromone. A stepped gradient of one or two intermediate light intensities could be implemented, but might be expected to have relatively little impact on swarm-level behaviour, given the limited controllability of the robots. Such a logic may also apply to nanoscale swarm systems.

Another aspect of our implementation is the continual production of pheromone trails by robots, regardless of whether they are already in a pheromone patch. If some form of intensity gradient was implemented in the real robots, they could 'choose' not to deposit pheromone in regions already marked. This may help to prevent trapping pheromone patches from forming. We investigated a threshold of 0.75 in simulation (electronic supplementary material, figures S7–S9 and tables S7–S9) and interestingly, the trend appears to be an improvement in initial dispersal (logistic growth rate $k$) but a decrease in longer-term performance ($L\_max$), especially in area coverage. Pheromone could also be laid such that it accumulates progressively rather than starting from full saturation. Without changing the robot or pheromone behaviour, the initial spatial positioning of the swarm in a square grid could also be adjusted to be more spread out to reduce the number of robots initially encircled by a pheromone patch. With such a change in initial conditions, one needs to consider the relationship with the envisaged real-world deployment scenario. An even simpler way of tackling the challenge of pheromone oversaturation could be to make the pheromone avoidance decision probabilistic (i.e. sometimes pheromone is ignored when detected). Alternatively, a swarm density-dependent controller that is able to adjust robot behaviour depending on the detected presence of neighbours, which may indeed vary considerably over time if robot 'clumping' occurs in response to pheromone, could be a useful direction in future swarm stigmergy research. This density-dependent behaviour could change either, or both, of the robot movement and decision to lay pheromone. For example, the negative consequences of high densities of army ants are managed through self-organized 'traffic lanes', which emerge in part from avoiding collisions before attending to pheromone information [4]. In the context of attractive recruitment pheromone trails to a food source, *Lasius niger* ants are observed to lay less pheromone in crowded conditions [54] (crowding negative feedback); while high levels of trail pheromone supresses further pheromone deposition [55] (pheromone negative feedback, as in the 0.75 threshold investigated here). Of these two effects, crowding negative feedback appears to be the more significant [56], suggesting the value of a swarm density-dependent pheromone-laying rule. Finally, the avoidance behaviour implemented by the robots in the results shown here was very simple, involving a turn to a random direction and then a period of forward motion (§3.2). One alternative approach would be to use a form of 'iterative deepening', whereby on pheromone detection a robot turns and travels a distance $k$, and $2*k$ ($n*k$) if pheromone is detected again ($n + 1$), requiring only one additional memory variable $n$. Our findings of stigmergy's limits under one avoidance approach do not, of course, mean that all approaches will be *necessarily* inferior to random walk at large $N$—though they do serve to highlight the engineering challenge of using this fundamental method of self-organization when environmental modifications saturate. This also challenges the general assumption that swarm strategies are scalable to large numbers.

In our exploration and area coverage scenario, stigmergy could be more effective when implemented for a time-limited period: for the initial dispersal into the environment, it may be valuable for reconnoitring all available space; and then once this initial phase is complete, stigmergy could be deactivated for very large swarms because non-interaction may be superior for area coverage and

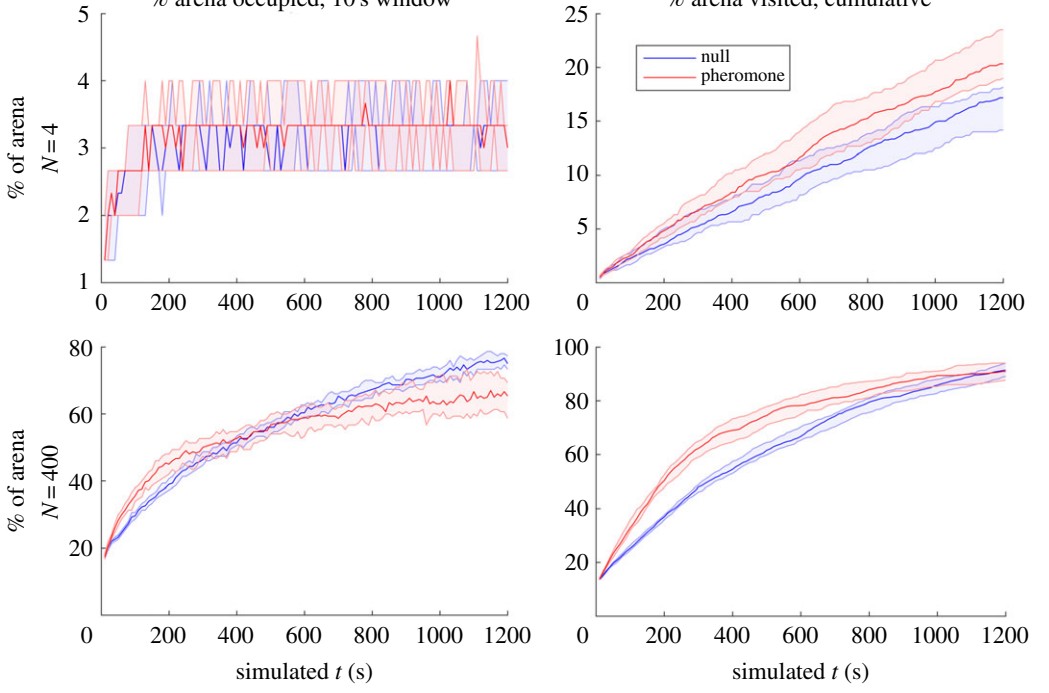

**Figure 4.** Performance of simulated robotic swarms ($N = 4$, 400; see electronic supplementary material for the complete set). Median and lower and upper quartiles shown for 30 simulations. For $N = 1$, there is no clear difference in performance in either metric, while at $N = 4$ onwards cumulative exploration is superior using stigmergy. At $N = 256$ upwards, area coverage becomes increasingly inferior to non-interacting robot swarms; and while collective exploration still shows an early lead, long-term performance is comparable to random walkers.

**Table 4.** Performance of simulated robotic swarms ($N = 64$, 324). Compare to table 3 (real robots).

| | $N = 64$ | | | | $N = 324$ | | | |
|---|---|---|---|---|---|---|---|---|
| | null | pher | $p$ (diff) | $t$ | null | pher | $p$ (diff) | $t$ |
| *area coverage* | | | | | | | | |
| L_max | 33.38 ± 2.16 | 36.57 ± 4.06 | <0.001 | −3.69 | 73.45 ± 3.65 | 62.46 ± 11.27 | <0.001 | 5.00 |
| rate $k$ | 0.005 ± 0.001 | 0.006 ± 0.002 | 0.018 | −2.45 | 0.003 ± 0.0004 | 0.007 ± 0.005 | <0.001 | −3.69 |
| *exploration* | | | | | | | | |
| L_max | 64.71 ± 6.34 | 77.23 ± 6.26 | <0.001 | −7.57 | 90.00 ± 5.03 | 85.61 ± 6.32 | 0.005 | 2.92 |
| rate $k$ | 0.004 ± 0.0004 | 0.005 ± 0.0008 | <0.001 | −4.48 | 0.004 ± 0.0003 | 0.007 ± 0.002 | <0.001 | −8.38 |

diffusing into all available space (though agents still interact inasmuch as they bump into each other). Alternatively, a subset of individuals could be pheromone layers and avoiders among a larger group of agents operating a very simple movement algorithm such as a random walk. Many natural systems that work in large numbers (cells, molecules, etc.) rely on reaction–diffusion, rather than complex communication, to achieve tasks. Perhaps random, uncoordinated motion is an effective, low-cost movement strategy at high agent densities: when a large $N$ swarm is introduced to a spatially constrained environment.

## 5. Conclusion

Social insects like ants are the archetypal inspiration for swarm robotics research: in particular, their use of pheromones facilitates highly effective, low-cost coordination of many individuals. Attractive or recruitment pheromones are well known in tasks like foraging, and though repellent or territory-

marking pheromones are less commonly employed, they are also recognized as highly effective for tasks such as area coverage and exploration. Yet repellent pheromones have only been tested in simulation and in robots for relatively small numbers ($N < 30$). Here, we demonstrated their use and effectiveness in swarms of Kilobots an order of magnitude larger ($N > 300$). However, we found that at higher swarm densities (larger $N$), such stigmergic coordination may become ineffective in certain respects relative to non-interacting swarm agents.

As with ants leaving chemical marker cues with their footprints, the robots in this study left pheromone on an ongoing basis. This is advantageous inasmuch as little or no processing activity needs to be expended on decision-making. However, as with the ants, there are pros and cons to a continuous passive signal. For high-density swarms, an additional cost may be recognized: the nullification or even harm to performance of stigmergic coordination because of environmental saturation with the social information (pheromone). This may be a particular challenge when the environment is spatially constrained, for example, by a physical barrier: and in cases like the repellent markers here, such constraints may be emergent and self-reinforcing via the pheromone itself. A higher pheromone decay constant (faster decay) appears to be more optimal in higher density (larger) groups, to compensate for the larger amount of pheromone being deposited. Faster decay (reduced social information sharing) may also be appropriate for dynamic environments rather than the static environment exploration considered here [57]. Yet, with an ever-shortening memory of movement trails, pheromone could become increasingly irrelevant to group-level behaviour. In ant colonies, it is unlikely that such large numbers of individuals would be exploring at once. Rather, such coordination via negative markers is thought to occur among species with smaller colony sizes, among a few scouts [58]. Thus, just as evolution has optimized the trade-off between swarm size and pheromone use for collective exploration, our study here points towards the limits of stigmergy at large $N$ swarm sizes in spatially constrained (high density) environments. Sometimes, 'less is more': less frequently laid pheromone, or fewer depositing agents as a subset of a larger group, may maximize stigmergy's usefulness. One optimizing step could be the use of hybrid swarms of many simple agents, with minimal processing power to carry out random walk-type behaviour, and a subset of individuals with pheromone-laying hardware and avoidance capabilities. Such swarms may be better equipped to achieve simultaneously good area coverage and exploration.

Our findings indicate that when working with a high-density robot swarm in a spatially constrained environment, more sophisticated swarm exploration and area coverage strategies may not bring advantages beyond that observed with random movement. Instead, for biological systems working at the subcellular scale, with very large numbers of 'agents', reaction–diffusion is relied upon for effective permeation of an environment [2]. Thus, although common design principles across length scales may exist for swarm engineering, swarm size (agent density) may be a key factor in determining the relevance of either stigmergic or diffusive movement rules.

Data accessibility. The data from the robot experiments, comprising overhead camera tracking of robot positions, and the simulated robot trajectories are available from Dryad Digital Repository: https://dx.doi.org/10.5061/dryad.bc4b0kk [59].

Authors' contributions. E.R.H. planned the study, created the robot tracking and projector pheromone system, conducted the experiments, analysed the data, and wrote the paper. S.J. created and extended the 'Kilobox' robot simulator and the Kilobot projector light colour detection script. S.H. advised on the project and helped to write the paper. All authors gave their final approval for publication.

Competing interests. We declare we have no competing interests.

Funding. E.R.H. thanks the UK Engineering and Physical Sciences Research Council (EPSRC DTP Doctoral Prize grant no. EP/N509619/1).

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
