## [Reviewer comments · Royal Society Open Science]

Review History

RSOS-190225.R0 (Original submission)

Review form: Reviewer 1

Is the manuscript scientifically sound in its present form?

Yes

Are the interpretations and conclusions justified by the results?

Yes

Is the language acceptable?

Yes

Is it clear how to access all supporting data?

Yes

Do you have any ethical concerns with this paper?

No

Have you any concerns about statistical analyses in this paper?

No

Recommendation?

Major revision is needed (please make suggestions in comments)

Comments to the Author(s)

%%% Overall comments: This is a very interesting paper that discusses the effects of using artificial pheromones to have a robotic swarm better cover and explore a bounded arena. The experiments are impressive with the deployment of 300+ Kilobots simultaneously. The results are extremely surprising considering the low robotic density in the arena but should be an extremely useful contribution if the results still hold when further expanded upon (details on the suggested expansion below).

%%% General Questions/Suggestions by Section

Overall

The authors mostly refer to the number of robots deployed. While it is an impressive number, should be listed along with the density which I think would be a more useful metric. The density should make it easier to compare these results with previous works.

I also wonder if the results of this paper hold and extend to situations where fewer numbers of robots are deployed in a smaller environment but the density is kept the same. Perhaps that could be explored in the simulation environment.

Introduction

When discussing the various implementation methods of pheromone trails by previous work, does the implementation matter? Does it effect the experimental results? It is unclear what the point the authors are making by breaking up the previous work in this manner if the main point at the end is the number of robots used.

Materials and Methods

I don't think the specific decay time for your experimental implementation is needed here.

Robot Controller

Why was the scattering behavior chosen this way? Does the 0.5s rotation rotate the robot 180 degrees? Why does is move forward for a randomly chosen time?

Do robots avoid boundaries? Is there any way to avoid buildup in corners/along walls or was this a non-issue?

Simulations

It is unclear why the parameters were chosen in this manner. How do these parameter choices effect the outcome?

Table 1: I am unsure what the point of showing the density in two different ways is or if it is necessary.

Results and Discussion

For figure 3 and 4 why does the simulation for $N = \text{Large}$ (324 and 400 respectively) seem different? Why does the pure random walk surpass the stigmergy approach in the simulation but not the robot experiments?

The authors should also discuss the rise time difference in the cumulative exploration plots for the real vs simulated experiment or at least give some intuition into what causes the difference seen in these results.

%% Result Expansion

I believe a lot of the results discussed here hinge on three major factors, the scattering implementation, initial conditions, and decay rate.

Scattering Controller:

Why do you choose 0.5 seconds for rotation but give a random forward progression for the scattering? Would the results change with a pure reflection controller (or as pure as the Kilobots can perform)? If the goal of the random forward motion is to have to robots "jump" the pheromone barrier sometimes, why not just give a probability to ignore/miss the pheromone detected?

Ideally, I would imagine a scattering controller would take agents away from the pheromone region in optimal time.

In general, the reason for this controller should be further explained and some intuition on these results to a general class of controller should be provided. For example, if controllers that drive the robots away from the detected pheromone in the direction they entered the region are used, the parameters that are applied to the Kilobots to make this happen should underscore the general controller not be the forefront.

Initial Conditions:

Why are the robots so tightly packed initially, wont this essentially trap the robots in the middle initially forcing these results? Figure 1 seems to show this. The robots seem trapped between the wall and initial middle starting zone with only a few escaping to explore.

Do these results hold if the robots are uniformly distributed through the arena to begin with or in a bimodal initial distribution?

Also is the initial heading of the Kilobots also randomized? Does that matter?

If the exploration has to start from an initial zone, maybe there is another way to begin the experiment such that robots aren't initially trapped.

Decay Rate:

An aside interesting experiment would be how the decay rate of the pheromones effects the exploration and coverage of the swarm. I wonder if there is an optimal decay time to maximize the amount explored.

In any case, it should be discussed why this decay rate/time was chosen for the experiments.

%%% Misc Comments

There are a few statements that I believe need citations. For example,

Lines 46 - 52 state there is a lot of research on the use of attractive pheromones but does not cite any of this previous research. Also claims there is research on repellent pheromones but does not provide a citation.

Some statements need further explanation:

Lines 120 - 122 the authors state that larger swarm sizes make pheromones hard to avoid. The way this is written leaves ambiguity, is it harder because of the higher density (i.e. robots have less motion options because there are other robots physically in the way), because of the increased amount of pheromone, or some other reason(s)?

%%% Small Writing Edits

Lines 43-45: The beginning of this sentence before the for example does not make sense to me. Should it read "has limits to its usefulness"?

Line 77: slime molds not slime melds.

Review form: Reviewer 2

Is the manuscript scientifically sound in its present form?

Yes

Are the interpretations and conclusions justified by the results?

No

Is the language acceptable?

Yes

Is it clear how to access all supporting data?

Yes

Do you have any ethical concerns with this paper?

No

Have you any concerns about statistical analyses in this paper?

No

Recommendation?

Major revision is needed (please make suggestions in comments)

Comments to the Author(s)

-- General comments --

This manuscript describes a study of a virtual repellent pheromone in the context of area coverage and exploration by a robotic swarm. The novelty of this study is that it was performed with a large number of robots (both in simulation and in reality), which allowed the authors to test the behavior of the swarm in crowded conditions. The main result of the study is that, as density increases, the repellent pheromone facilitates less and less the exploration and dispersion of the robots in the experimental arena when compared to non-interacting robots. At very high density, it even becomes detrimental to the dispersion of the robots.

The manuscript is well written and easy to follow. All the steps to reproduce the work are described and the data from the experiments are provided. However, reproducibility could be improved by providing the code for the controller of the robots, the simulations and the data analysis (see journal policy on data and code sharing at <https://royalsociety.org/journals/ethics-policies/data-sharing-mining/>).

Overall I find this study interesting but somewhat limited in scope. The negative consequences of high densities on self-organizing behaviors has been very well studied, in particular in the context of traffic and evacuation (including in combination with stigmergy in studies of route selection in ants for instance). In particular, it has often been found that the scale of these negative consequences can be very sensitive to small variations in the behavioral parameters of the system (and for combinations of parameters could even be cancelled). In the present study, the authors did not manipulate a single behavioral parameter in order to explore the sensitivity of their result to the behavior of the robots. As a consequence, it is difficult to assess how general the conclusions of the study are since they are limited to comparing one set of behavioral parameters to the control group. While I completely understand that such a sensitivity analysis cannot be performed with the real robots, it could have been easily done in the simulations. In particular, the authors could have tested the influence of the following parameters on exploration and coverage:

- Detection threshold; the threshold is fixed at 0.5. Why not measure the sensitivity of the results to this arbitrarily set parameter?
- Evaporation rate; same question as before.
- Pheromone accumulation; if I understand correctly, every act of pheromone deposition reset the pheromone intensity to its maximum value. Why not testing with progressive accumulation instead?
- Pheromone deposition; in the discussion, the authors hint at the idea of allowing deposition at a location only if the pheromone concentration is below a certain concentration (as certain ants do). Why not test its effect since it seems a trivial modification of the pheromone deposition code.
- Arena size; in the current experiment, the arena size is fixed, so it's not possible to determine whether the observed effect is due to an increase in density or an increase in population size. To decouple them, the authors could repeat the analysis with different arena sizes (e.g., comparing different arena sizes while maintaining the density constant).

My general recommendation is to ask the authors to perform at least some of this additional sensitivity analysis with their simulation tool before resubmitting their manuscript. Without it,

the scope of the results is very limited and I fail to see how it contributes significantly to the field of swarm robotics.

-- Minor comments --

l. 77: "slime melds"  "slime molds".

l. 103: and also in "Campo, A., Gutiérrez, A., Nouyan, S., Pinciroli, C., Longchamp, V., Garnier, S., et al. (2010). Artificial pheromone for path selection by a foraging swarm of robots. *Biol. Cybern.* 103, 339–352. doi:10.1007/s00422-010-0402-x".

l. 112: N=10 in "Garnier, S., Combe, M., Jost, C., and Theraulaz, G. (2013). Do ants need to estimate the geometrical properties of trail bifurcations to find an efficient route? A swarm robotics test bed. *PLoS Comput. Biol.* 9, e1002903. doi:10.1371/journal.pcbi.1002903".

Fig. 2: please indicate significance on the graph.

l. 289-290: please add the effect size for both cases since your conclusion here is about the reduction in the difference between random and stigmergy (the difference is significant in both cases).

Fig. 3-4 (and corresponding text in the body of manuscript): the analysis of these results is purely qualitative and could be improved to make a more convincing point. For instance, given the form of the graphs, the authors could fit a logistic curve to the data from each replicate and extract their growth rates and maximum values. They could then formally compare these two descriptors of the dynamics of the swarm across the different experimental treatments using appropriate statistical tests.

Decision letter (RSOS-190225.R0)

10-May-2019

Dear Dr Hunt,

The editors assigned to your paper ("Testing the limits of pheromone stigmergy in spatially constrained robotic swarms") have now received comments from reviewers. We would like you to revise your paper in accordance with the referee and Associate Editor suggestions which can be found below (not including confidential reports to the Editor). Please note this decision does not guarantee eventual acceptance.

Please submit a copy of your revised paper before 02-Jun-2019. Please note that the revision deadline will expire at 00.00am on this date. If we do not hear from you within this time then it will be assumed that the paper has been withdrawn. In exceptional circumstances, extensions may be possible if agreed with the Editorial Office in advance. We do not allow multiple rounds of revision so we urge you to make every effort to fully address all of the comments at this stage. If deemed necessary by the Editors, your manuscript will be sent back to one or more of the original reviewers for assessment. If the original reviewers are not available, we may invite new reviewers.

- Data accessibility

If you wish to submit your supporting data or code to Dryad (<http://datadryad.org/>), or modify your current submission to dryad, please use the following link:
<http://datadryad.org/submit?journalID=RSOS&manu=RSOS-190225>

- Competing interests

- Authors' contributions

- Acknowledgements

- Funding statement

Kind regards,

Andrew Dunn

on behalf of Professor Brooke Flammang (Associate Editor) and R. Kerry Rowe (Subject Editor)
openscience@royalsociety.org

Comments to Author:

Reviewers' Comments to Author:

Reviewer: 1

Comments to the Author(s)

%%% Overall comments: This is a very interesting paper that discusses the effects of using artificial pheromones to have a robotic swarm better cover and explore a bounded arena. The experiments are impressive with the deployment of 300+ Kilobots simultaneously. The results are extremely surprising considering the low robotic density in the arena but should be an extremely useful contribution if the results still hold when further expanded upon (details on the suggested expansion below).

%%% General Questions/Suggestions by Section

Overall

The authors mostly refer to the number of robots deployed. While it is an impressive number, should be listed along with the density which I think would be a more useful metric. The density should make it easier to compare these results with previous works.

I also wonder if the results of this paper hold and extend to situations where fewer numbers of robots are deployed in a smaller environment but the density is kept the same. Perhaps that could be explored in the simulation environment.

Introduction

When discussing the various implementation methods of pheromone trails by previous work, does the implementation matter? Does it effect the experimental results? It is unclear what the point the authors are making by breaking up the previous work in this manner if the main point at the end is the number of robots used.

Materials and Methods

I don't think the specific decay time for your experimental implementation is needed here.

Robot Controller

Why was the scattering behavior chosen this way? Does the 0.5s rotation rotate the robot 180 degrees? Why does it move forward for a randomly chosen time?

Do robots avoid boundaries? Is there any way to avoid buildup in corners/along walls or was this a non-issue?

Simulations

It is unclear why the parameters were chosen in this manner. How do these parameter choices effect the outcome?

Table 1: I am unsure what the point of showing the density in two different ways is or if it is necessary.

Results and Discussion

For figure 3 and 4 why does the simulation for $N = \text{Large}$ (324 and 400 respectively) seem different? Why does the pure random walk surpass the stigmergy approach in the simulation but not the robot experiments?

The authors should also discuss the rise time difference in the cumulative exploration plots for the real vs simulated experiment or at least give some intuition into what causes the difference seen in these results.

%% Result Expansion

I believe a lot of the results discussed here hinge on three major factors, the scattering implementation, initial conditions, and decay rate.

Scattering Controller:

Why do you choose 0.5 seconds for rotation but give a random forward progression for the scattering? Would the results change with a pure reflection controller (or as pure as the Kilobots can perform)? If the goal of the random forward motion is to have to robots "jump" the

pheromone barrier sometimes, why not just give a probability to ignore/miss the pheromone detected?

Ideally, I would imagine a scattering controller would take agents away from the pheromone region in optimal time.

In general, the reason for this controller should be further explained and some intuition on these results to a general class of controller should be provided. For example, if controllers that drive the robots away from the detected pheromone in the direction they entered the region are used, the parameters that are applied to the Kilobots to make this happen should underscore the general controller not be the forefront.

Initial Conditions:

Why are the robots so tightly packed initially, wont this essentially trap the robots in the middle initially forcing these results? Figure 1 seems to show this. The robots seem trapped between the wall and initial middle starting zone with only a few escaping to explore.

Do these results hold if the robots are uniformly distributed through the arena to begin with or in a bimodal initial distribution?

Also is the initial heading of the Kilobots also randomized? Does that matter?

If the exploration has to start from an initial zone, maybe there is another way to begin the experiment such that robots aren't initially trapped.

Decay Rate:

An aside interesting experiment would be how the decay rate of the pheromones effects the exploration and coverage of the swarm. I wonder if there is an optimal decay time to maximize the amount explored.

In any case, it should be discussed why this decay rate/time was chosen for the experiments.

%%% Misc Comments

There are a few statements that I believe need citations. For example,

Lines 46 - 52 state there is a lot of research on the use of attractive pheromones but does not cite any of this previous research. Also claims there is research on repellent pheromones but does not provide a citation.

Some statements need further explanation:

Lines 120 - 122 the authors state that larger swarm sizes make pheromones hard to avoid. The way this is written leaves ambiguity, is it harder because of the higher density (i.e. robots have less motion options because there are other robots physically in the way), because of the increased amount of pheromone, or some other reason(s)?

%%% Small Writing Edits

Lines 43-45: The beginning of this sentence before the for example does not make sense to me. Should it read "has limits to its usefulness"?

Line 77: slime molds not slime melds.

Reviewer: 2

Comments to the Author(s)

-- General comments --

This manuscript describes a study of a virtual repellent pheromone in the context of area coverage and exploration by a robotic swarm. The novelty of this study is that it was performed with a large number of robots (both in simulation and in reality), which allowed the authors to test the behavior of the swarm in crowded conditions. The main result of the study is that, as density increases, the repellent pheromone facilitates less and less the exploration and dispersion of the robots in the experimental arena when compared to non-interacting robots. At very high density, it even becomes detrimental to the dispersion of the robots.

The manuscript is well written and easy to follow. All the steps to reproduce the work are described and the data from the experiments are provided. However, reproducibility could be improved by providing the code for the controller of the robots, the simulations and the data analysis (see journal policy on data and code sharing at <https://royalsociety.org/journals/ethics-policies/data-sharing-mining/>).

Overall I find this study interesting but somewhat limited in scope. The negative consequences of high densities on self-organizing behaviors has been very well studied, in particular in the context of traffic and evacuation (including in combination with stigmergy in studies of route selection in ants for instance). In particular, it has often been found that the scale of these negative consequences can be very sensitive to small variations in the behavioral parameters of the system (and for combinations of parameters could even be cancelled). In the present study, the authors did not manipulate a single behavioral parameter in order to explore the sensitivity of their result to the behavior of the robots. As a consequence, it is difficult to assess how general the conclusions of the study are since they are limited to comparing one set of behavioral parameters to the control group. While I completely understand that such a sensitivity analysis cannot be performed with the real robots, it could have been easily done in the simulations. In particular, the authors could have tested the influence of the following parameters on exploration and coverage:

- Detection threshold; the threshold is fixed at 0.5. Why not measure the sensitivity of the results to this arbitrarily set parameter?
- Evaporation rate; same question as before.
- Pheromone accumulation; if I understand correctly, every act of pheromone deposition reset the pheromone intensity to its maximum value. Why not testing with progressive accumulation instead?
- Pheromone deposition; in the discussion, the authors hint at the idea of allowing deposition at a location only if the pheromone concentration is below a certain concentration (as certain ants do). Why not test its effect since it seems a trivial modification of the pheromone deposition code.
- Arena size; in the current experiment, the arena size is fixed, so it's not possible to determine whether the observed effect is due to an increase in density or an increase in population size. To

decouple them, the authors could repeat the analysis with different arena sizes (e.g., comparing different arena sizes while maintaining the density constant).

My general recommendation is to ask the authors to perform at least some of this additional sensitivity analysis with their simulation tool before resubmitting their manuscript. Without it, the scope of the results is very limited and I fail to see how it contributes significantly to the field of swarm robotics.

-- Minor comments --

l. 77: "slime melds"  "slime molds".

l. 103: and also in "Campo, A., Gutiérrez, A., Nouyan, S., Pinciroli, C., Longchamp, V., Garnier, S., et al. (2010). Artificial pheromone for path selection by a foraging swarm of robots. *Biol. Cybern.* 103, 339–352. doi:10.1007/s00422-010-0402-x".

l. 112: N=10 in "Garnier, S., Combe, M., Jost, C., and Theraulaz, G. (2013). Do ants need to estimate the geometrical properties of trail bifurcations to find an efficient route? A swarm robotics test bed. *PLoS Comput. Biol.* 9, e1002903. doi:10.1371/journal.pcbi.1002903".

Fig. 2: please indicate significance on the graph.

l. 289-290: please add the effect size for both cases since your conclusion here is about the reduction in the difference between random and stigmergy (the difference is significant in both cases).

Fig. 3-4 (and corresponding text in the body of manuscript): the analysis of these results is purely qualitative and could be improved to make a more convincing point. For instance, given the form of the graphs, the authors could fit a logistic curve to the data from each replicate and extract their growth rates and maximum values. They could then formally compare these two descriptors of the dynamics of the swarm across the different experimental treatments using appropriate statistical tests.

Author's Response to Decision Letter for (RSOS-190225.R0)

See Appendix A.

RSOS-190225.R1 (Revision)

Review form: Reviewer 1

Is the manuscript scientifically sound in its present form?

Yes

Are the interpretations and conclusions justified by the results?

Yes

Is the language acceptable?

Yes

Do you have any ethical concerns with this paper?

No

Have you any concerns about statistical analyses in this paper?

No

Recommendation?

Accept with minor revision (please list in comments)

Comments to the Author(s)

Overall Review:

The authors have addressed most of my previous concerns. I remain curious about the initialization of the swarm which I explain below. However, the reason for the initialization has been addressed in this updated draft. Overall, the mild concern I had does not detract from the contribution of the paper and I believe it is suitable for publication.

I do have minor suggestions for this new draft that may improve the paper. I will list them below in the order they appear in the paper.

Initial condition concern from previous review and response:

Why are the robots so tightly packed initially, wont this essentially trap the robots in the middle initially forcing these results?

>> We would disagree that they are packed tightly, this would seem to imply they start in physical contact when in fact they have 7mm spacing (Methods 3.1).

In the context of an exploration or area coverage task, the scenario we have in mind is that they would be deposited in one location by a user, to then perform the job of spreading out.

To clarify, I did not mean to imply tightly packed was such that the robots could not move but that the initial pheromone deposition of the outer most agents would create a container for the inner robots (inner robots can only escape through 7mm gaps if none of the outside robots closed those gaps further. This may not have been a problem but hopefully clarifies my concern.

I do not think it is necessary to include another set of experiments but I still am interested if a different initial condition (long rectangle in stead of a square) would effect these results (as less agents would be in the center of the formation).

Minor suggestions

Table 1: I am not a fan of "Here" as the reference for this paper, especially with the bottom ref cell being left blank. Maybe reformat the table with a top section for this paper's entries split with a double horizontal line for comparison to other papers.

This is a nit-picky suggestion. This table is a great addition to the paper as is!

Line 279-280, the math here seems out of place and unnecessary. I think you could say in a rectangular arena half the area of the original with same aspect ratio (or something to that regard).

Section 3.3.2 - These are great but the results are in the supplementary material. When I read this into the discussion section I was unclear what parameter set was being used for the following tables/figures. It should be made clear here or in the beginning of Section 4 that the results in the main manuscript are for the default scenario with the others in the supplementary material.

Table 3/4 - The formatting of this table should be changed to make it visually cleaner and easier to read

Review form: Reviewer 2

Is the manuscript scientifically sound in its present form?

Yes

Are the interpretations and conclusions justified by the results?

Yes

Is the language acceptable?

Yes

Do you have any ethical concerns with this paper?

No

Have you any concerns about statistical analyses in this paper?

No

Recommendation?

Accept as is

Comments to the Author(s)

The manuscript is much improved since its first version and I believe it is now ready for publication. Thanks very much for taking the time to include the suggestions of both reviewers. I think it makes for a stronger case for your hypothesis.

Decision letter (RSOS-190225.R1)

08-Sep-2019

Dear Dr Hunt:

On behalf of the Editors, I am pleased to inform you that your Manuscript RSOS-190225.R1

entitled "Testing the limits of pheromone stigmergy in high density robot swarms" has been accepted for publication in Royal Society Open Science subject to minor revision in accordance with the referee suggestions. Please find the referees' comments at the end of this email.

The reviewers and Subject Editor have recommended publication, but also suggest some minor revisions to your manuscript. Therefore, I invite you to respond to the comments and revise your manuscript.

- Ethics statement

- Data accessibility

If you wish to submit your supporting data or code to Dryad (<http://datadryad.org/>), or modify your current submission to dryad, please use the following link:
<http://datadryad.org/submit?journalID=RSOS&manu=RSOS-190225.R1>

- Competing interests

- Authors' contributions

- Acknowledgements

- Funding statement

Because the schedule for publication is very tight, it is a condition of publication that you submit the revised version of your manuscript before 17-Sep-2019. Please note that the revision deadline will expire at 00.00am on this date. If you do not think you will be able to meet this date please let me know immediately.

on behalf of Professor Brooke Flammang (Associate Editor) and R. Kerry Rowe (Subject Editor)
 openscience@royalsociety.org

Associate Editor Comments to Author (Professor Brooke Flammang):

Associate Editor: 1

Comments to the Author:

In general both reviewers were satisfied with this manuscript in its current form. While reviewer 1 did list a couple additional corrections that would be useful, one in particular should be attended to:

"Section 3.3.2 - These are great but the results are in the supplementary material. When I read this into the discussion section I was unclear what parameter set was being used for the following tables/figures. It should be made clear here or in the beginning of Section 4 that the results in the main manuscript are for the default scenario with the others in the supplementary material."

Reviewer comments to Author:

Reviewer: 1

Comments to the Author(s)

Overall Review:

The authors have addressed most of my previous concerns. I remain curious about the initialization of the swarm which I explain below. However, the reason for the initialization has been addressed in this updated draft. Overall, the mild concern I had does not detract from the contribution of the paper and I believe it is suitable for publication.

I do have minor suggestions for this new draft that may improve the paper. I will list them below in the order they appear in the paper.

 Initial condition concern from previous review and response:

Why are the robots so tightly packed initially, wont this essentially trap the robots in the middle initially forcing these results?

>> We would disagree that they are packed tightly, this would seem to imply they start in physical contact when in fact they have 7mm spacing (Methods 3.1).

In the context of an exploration or area coverage task, the scenario we have in mind is that they would be deposited in one location by a user, to then perform the job of spreading out.

To clarify, I did not mean to imply tightly packed was such that the robots could not move but that the initial pheromone deposition of the outer most agents would create a container for the inner robots (inner robots can only escape through 7mm gaps if none of the outside robots closed those gaps further. This may not have been a problem but hopefully clarifies my concern.

I do not think it is necessary to include another set of experiments but I still am interested if a different initial condition (long rectangle in stead of a square) would effect these results (as less agents would be in the center of the formation).

 Minor suggestions

Table 1: I am not a fan of "Here" as the reference for this paper, especially with the bottom ref cell being left blank. Maybe reformat the table with a top section for this paper's entries split with a double horizontal line for comparison to other papers.

This is a nit-picky suggestion. This table is a great addition to the paper as is!

Line 279-280, the math here seems out of place and unnecessary. I think you could say in a rectangular arena half the area of the original with same aspect ratio (or something to that regard).

Section 3.3.2 - These are great but the results are in the supplementary material. When I read this into the discussion section I was unclear what parameter set was being used for the following tables/figures. It should be made clear here or in the beginning of Section 4 that the results in the main manuscript are for the default scenario with the others in the supplementary material.

Table 3/4 - The formatting of this table should be changed to make it visually cleaner and easier to read

Reviewer: 2

Comments to the Author(s)

The manuscript is much improved since its first version and I believe it is now ready for publication. Thanks very much for taking the time to include the suggestions of both reviewers. I think it makes for a stronger case for your hypothesis.

Author's Response to Decision Letter for (RSOS-190225.R1)

See Appendix B.

Decision letter (RSOS-190225.R2)

20-Sep-2019

Dear Dr Hunt,

I am pleased to inform you that your manuscript entitled "Testing the limits of pheromone stigmergy in high density robot swarms" is now accepted for publication in Royal Society Open Science.

on behalf of Professor Brooke Flammang (Associate Editor) and R. Kerry Rowe (Subject Editor)
openscience@royalsociety.org

Appendix A

Response to reviewers – “Testing the limits of pheromone stigmergy in spatially constrained robotic swarms”

We thank both of the reviewers for their insightful and constructive reviews. Please see below a point-by-point response to their feedback in blue text.

Comments to Author:

Reviewers' Comments to Author:

Reviewer: 1

Comments to the Author(s)

%%% Overall comments: This is a very interesting paper that discusses the effects of using artificial pheromones to have a robotic swarm better cover and explore a bounded arena. The experiments are impressive with the deployment of 300+ Kilobots simultaneously. The results are extremely surprising considering the low robotic density in the arena but should be an extremely useful contribution if the results still hold when further expanded upon (details on the suggested expansion below).

We are pleased the reviewer found the paper very interesting, thank you for your helpful feedback.

%%% General Questions/Suggestions by Section

Overall

The authors mostly refer to the number of robots deployed. While it is an impressive number, should be listed along with the density which I think would be a more useful metric. The density should make it easier to compare these results with previous works.

>> Thank you for this suggestion, we have added a new Table 1 which compares the density of some previously researched robot swarms with the present paper.

I also wonder if the results of this paper hold and extend to situations where fewer numbers of robots are deployed in a smaller environment but the density is kept the same. Perhaps that could be explored in the simulation environment.

>> Thank you for this helpful suggestion for further simulation work. Following this feedback, also from Reviewer 2, we have made these additional investigations:

1. We have repeated the analysis with a smaller arena with a smaller swarm size, hence varying the number of robots while maintaining the swarm density.
2. We have repeated the analysis with one higher, and one lower pheromone decay rate.
3. We have repeated the analysis with a pheromone saturation threshold, such that further pheromone is only laid on an area when below a threshold of 0.75 (1=full saturation).
4. We have made additional, quantitative assessment of swarm performance by fitting logistic curves and comparing the growth rate and maximum value parameters.

Introduction

When discussing the various implementation methods of pheromone trails by previous

work, does the implementation matter? Does it effect the experimental results? It is unclear what the point the authors are making by breaking up the previous work in this manner if the main point at the end is the number of robots used.

>> This section was meant as a methodological overview – it's an interesting point whether the implementation affects the behaviours observed. We now add a point on this to begin the section.

Materials and Methods

I don't think the specific decay time for your experimental implementation is needed here.

>> Including this detail is now better motivated, because as shown in the new simulations decay time does impact the behaviour of the swarm

Robot Controller

Why was the scattering behavior chosen this way? Does the 0.5s rotation rotate the robot 180 degrees? Why does it move forward for a randomly chosen time?

>> It is not possible to control Kilobot movements precisely. This scattering method was chosen as being a simple behaviour that is sufficient to deflect them from their current path, and can be compared to run-and-tumble movement dynamics in bacteria. The random period of forward motion helps them deal with boundaries, which they are not aware of (see answer below).

This has been updated in the text, in the Methods and also mentioned in the Results.

Do robots avoid boundaries? Is there any way to avoid buildup in corners/along walls or was this a non-issue?

The Kilobots do not have boundary proximity detectors – they are purposefully a very simple design. As such, they do not know when they have hit an obstacle.

At the end of the experiment the majority of the robots are not touching the walls. The random motion helps to move them away from walls if they are in that position. We have clarified this in the first paragraph of section 3.2.

Simulations

It is unclear why the parameters were chosen in this manner. How do these parameter choices effect the outcome?

>> We have performed additional simulations to explore the effect of parameter changes. The decay constant was chosen to ensure that the pheromone was present for long enough in the environment to have a clear effect on the robot behaviour, a factor which is relative to the rather slow movement speed of the robots. See also the answer to Reviewer 2.

Table 1: I am unsure what the point of showing the density in two different ways is or if it is necessary.

>> We think that number of robots per area and proportion of arena covered are indeed

two different characteristics which could be of interest to later researchers. The latter metric will depend on the type of robot deployed.

Results and Discussion

For figure 3 and 4 why does the simulation for $N = \text{Large}$ (324 and 400 respectively) seem different?

>> The main message here is that in large swarms, pheromone may no longer be helpful to the swarm's goal; we are validating that here in simulation and reality. Qualitatively the graphs are fairly comparable; for robot swarms the level of reality gap is actually quite good.

Why does the pure random walk surpass the stigmergy approach in the simulation but not the robot experiments?

>> We now acknowledge this in the Results section, around lines 406-414.

The authors should also discuss the rise time difference in the cumulative exploration plots for the real vs simulated experiment or at least give some intuition into what causes the difference seen in these results.

>> Thank you for this suggestion, we now discuss this factor in new sentence(s) on lines: 414-418.

%%% Result Expansion

I believe a lot of the results discussed here hinge on three major factors, the scattering implementation, initial conditions, and decay rate.

Scattering Controller:

Why do you choose 0.5 seconds for rotation but give a random forward progression for the scattering?

>> See our earlier answer to 'Robot Controller'.

Would the results change with a pure reflection controller (or as pure as the Kilobots can perform)?

>> It's very difficult to perform a 180-degree reflection on the Kilobot. Because they do not have localisation capabilities the needed incident trajectory is unknown. In practice it can deflect through a curved trajectory.

If the goal of the random forward motion is to have to robots "jump" the pheromone barrier sometimes, why not just give a probability to ignore/miss the pheromone detected?

>> This is a good suggestion for future work which we now mention in the Discussion lines 474-476.

Indeed, many swarms implementing stigmergy would not use this approach, and so our purpose here is to help take forward thinking about how performance of dense swarms can be optimised.

Ideally, I would imagine a scattering controller would take agents away from the pheromone region in optimal time.

>> The notion of an 'optimal' controller is interesting, because while it might be optimal for low density might underperform at higher density.

It seems that finding the optimal scattering controller would be a different research question than the one considered in the present paper. We demonstrate in Figure 2 that indeed, the avoidance behaviour does result in less time spent in the pheromone overall.

A robot density responsive controller would be something useful for the future - we have added to the Discussion (lines 476-479) the opportunities for future work to retain the relevance of stigmergy in non-uniform density environments.

In general, the reason for this controller should be further explained and some intuition on these results to a general class of controller should be provided.

For example, if controllers that drive the robots away from the detected pheromone in the direction they entered the region are used, the parameters that are applied to the Kilobots to make this happen should underscore the general controller not be the forefront.

>> In the context of swarm robotics it would seem like Kilobots are particularly simple in their repertoire of behaviours, and so a general class of controller would still seem to be quite specific to their domain.

Also see our earlier answer on the optimal controller. We have better motivated our choice of algorithm in relation to biology and other pheromone robotics, which seems a good entry point to amplify thoughts about how stigmergy interacts with group size and density.

Initial Conditions:

Why are the robots so tightly packed initially, wont this essentially trap the robots in the middle initially forcing these results?

>> We would disagree that they are packed tightly, this would seem to imply they start in physical contact when in fact they have 7mm spacing (Methods 3.1).

In the context of an exploration or area coverage task, the scenario we have in mind is that they would be deposited in one location by a user, to then perform the job of spreading out.

Figure 1 seems to show this. The robots seem trapped between the wall and initial middle starting zone with only a few escaping to explore.

>> We started the swarm in the centre of the arena using a projected guidance square, so that robots had the same starting position each time. Figure 1 shows them clustered some way off from the centre.

Do these results hold if the robots are uniformly distributed through the arena to begin with or in a bimodal initial distribution?

>> Starting the robots uniformly distributed through the arena would seem to pre-empt the goal of the task, which is indeed to spread out uniformly.

Also is the initial heading of the Kilobots also randomized? Does that matter?

>> Yes, it is randomised in simulation and as far as possible in the physical trials. One would expect there to be some initial effect if, for example, all the Kilobots were orientated in one direction, though the 'random walk' type behaviour of the robots might indeed wash this out after a period of time. We now clarify this in the Methods section, lines 159-160.

If the exploration has to start from an initial zone, maybe there is another way to begin the experiment such that robots aren't initially trapped.

>> As we mentioned, we do not think the robots are trapped – and indeed if the goal of the experiment is to explore the effects of very large group sizes on an area coverage and exploration task, we think some sort of initial clustering is unavoidable. See new remark lines 157-159.

Decay Rate:

An aside interesting experiment would be how the decay rate of the pheromones affects the exploration and coverage of the swarm. I wonder if there is an optimal decay time to maximize the amount explored.

>> We now perform new analysis of decay rate – e.g. lines 432-437.

In any case, it should be discussed why this decay rate/time was chosen for the experiments.

>> We have added an extra sentence on the choice of decay rate on lines 189-191.

%%% Misc Comments

There are a few statements that I believe need citations. For example,

Lines 46 - 52 state there is a lot of research on the use of attractive pheromones but does not cite any of this previous research.

We have added citations.

Also claims there is research on repellent pheromones but does not provide a citation.

>> Section 2 does go on to give an overview of this research, but we have added a few citations again at this point in the Introduction as you suggest.

Some statements need further explanation:

Lines 120 - 122 the authors state that larger swarm sizes make pheromones hard to avoid. The way this is written leaves ambiguity, is it harder because of the higher density (i.e. robots have less motion options because there are other robots physically in the way), because of the increased amount of pheromone, or some other reason(s)?

>> Thank you for pointing this out, we have added clarification that high swarm density leading to large amounts of pheromone is our main focus. See also the adjusted paper title.

%%% Small Writing Edits

Lines 43-45: The beginning of this sentence before the for example does not make sense

to me. Should it read "has limits to its usefulness"?

>> We have changed to 'has limits' as you suggest.

Line 77: slime molds not slime melds.

>> Fixed.

Reviewer: 2

Comments to the Author(s)

-- General comments --

This manuscript describes a study of a virtual repellent pheromone in the context of area coverage and exploration by a robotic swarm. The novelty of this study is that it was performed with a large number of robots (both in simulation and in reality), which allowed the authors to test the behavior of the swarm in crowded conditions. The main result of the study is that, as density increases, the repellent pheromone facilitates less and less the exploration and dispersion of the robots in the experimental arena when compared to non-interacting robots. At very high density, it even becomes detrimental to the dispersion of the robots.

The manuscript is well written and easy to follow. All the steps to reproduce the work are described and the data from the experiments are provided.

>> Thank you, we are pleased you have appreciated the results of the study.

However, reproducibility could be improved by providing the code for the controller of the robots, the simulations and the data analysis (see journal policy on data and code sharing at <https://royalsociety.org/journals/ethics-policies/data-sharing-mining/>).

>> We have included a new ESM 3 with code.

Overall I find this study interesting but somewhat limited in scope.

>> We are glad you find the study interesting – and have broadened the scope by following your recommendation to perform more simulations.

The negative consequences of high densities on self-organizing behaviors has been very well studied, in particular in the context of traffic and evacuation (including in combination with stigmergy in studies of route selection in ants for instance).

>> This may indeed be the case in the context of biology and pedestrian behaviour, but to our knowledge there are not many studies of this phenomenon in the robotics context. The focus is not density per se but stigmergy at higher density; given the assumption that stigmergy is a suitable way to coordinate large numbers in swarm robotics, we are probing that assumption.

In particular, it has often been found that the scale of these negative consequences can be very sensitive to small variations in the behavioral parameters of the system (and for combinations of parameters could even be cancelled).

>> We would be happy to include further citations if the reviewer would like to suggest them.

In the present study, the authors did not manipulate a single behavioral parameter in order to explore the sensitivity of their result to the behavior of the robots. As a

consequence, it is difficult to assess how general the conclusions of the study are since they are limited to comparing one set of behavioral parameters to the control group. While I completely understand that such a sensitivity analysis cannot be performed with the real robots, it could have been easily done in the simulations.

>> Thank you for this point. We have now performed additional simulations in response to your specific suggestions below.

In particular, the authors could have tested the influence of the following parameters on exploration and coverage:

- Detection threshold; the threshold is fixed at 0.5. Why not measure the sensitivity of the results to this arbitrarily set parameter?

>> We do not think that the detection threshold is significant considered separately from the evaporation rate – it would seem to be essentially adjusting the effective rate, which we do now investigate (see below).

- Evaporation rate; same question as before.

>> The main results were presented with a decay constant of 0.005, which resulted in a decay below the 0.5 threshold in around 140s (138.6s)

To perform an initial sensitivity analysis of the results to this parameter we examine a decay time of half (69.3s) and double (277.2s) this, which corresponds to decay constants of 0.01 and 0.0025.

- Pheromone accumulation; if I understand correctly, every act of pheromone deposition reset the pheromone intensity to its maximum value. Why not testing with progressive accumulation instead?

>> Given that our setup with the real robots used binary pheromone in the implementation, we felt this was an intuitive starting point. We have mentioned the potential to use progressive accumulation in the Discussion line 491.

- Pheromone deposition; in the discussion, the authors hint at the idea of allowing deposition at a location only if the pheromone concentration is below a certain concentration (as certain ants do). Why not test its effect since it seems a trivial modification of the pheromone deposition code.

>> We have performed this analysis for a threshold of 0.75.

- Arena size; in the current experiment, the arena size is fixed, so it's not possible to determine whether the observed effect is due to an increase in density or an increase in population size. To decouple them, the authors could repeat the analysis with different arena sizes (e.g., comparing different arena sizes while maintaining the density constant).

>> We have repeated the analysis in simulation with a smaller arena and smaller group size (i.e. maintaining density as you suggest).

My general recommendation is to ask the authors to perform at least some of this additional sensitivity analysis with their simulation tool before resubmitting their manuscript. Without it, the scope of the results is very limited and I fail to see how it contributes significantly to the field of swarm robotics.

>> Thank you for this encouragement – our additional analysis with 2 more decay constants, additional arena size, and deposition threshold, has confirmed our qualitative finding that stigmergy is more useful only in certain contexts.

-- Minor comments --

l. 77: "slime melds"  "slime molds".

>> Fixed.

l. 103: and also in "Campo, A., Gutiérrez, A., Nouyan, S., Pinciroli, C., Longchamp, V., Garnier, S., et al. (2010). Artificial pheromone for path selection by a foraging swarm of robots. *Biol. Cybern.* 103, 339–352. doi:10.1007/s00422-010-0402-x".

>> Now cited, thank you.

l. 112: N=10 in "Garnier, S., Combe, M., Jost, C., and Theraulaz, G. (2013). Do ants need to estimate the geometrical properties of trail bifurcations to find an efficient route? A swarm robotics test bed. *PLoS Comput. Biol.* 9, e1002903. doi:10.1371/journal.pcbi.1002903".

>> Now cited, thank you.

Fig. 2: please indicate significance on the graph.

>> Done.

l. 289-290: please add the effect size for both cases since your conclusion here is about the reduction in the difference between random and stigmergy (the difference is significant in both cases).

>> Now included.

Fig. 3-4 (and corresponding text in the body of manuscript): the analysis of these results is purely qualitative and could be improved to make a more convincing point. For instance, given the form of the graphs, the authors could fit a logistic curve to the data from each replicate and extract their growth rates and maximum values. They could then formally compare these two descriptors of the dynamics of the swarm across the different experimental treatments using appropriate statistical tests.

>> Thank you for this good suggestion, we have now performed this analysis for all the swarm performance data presented.

Appendix B

Response to reviewers

We thank both of the reviewers and the associate editor for this final round of review. Please see below a point-by-point response to feedback in blue text.

In addition, we have added some further references on density-dependent swarm performance to lines 76-81 and lines 115-130 and density-dependent pheromone laying to lines 629-637; another pheromone robotics study to lines 207-209 and Table 1 (ref [32], Llenas et al.); and mentioned the bearing of social information in dynamic environments (lines 682-683, Pitonakova et al. ref [58]).

Associate Editor Comments to Author (Professor Brooke Flammang):

Associate Editor: 1

Comments to the Author:

In general both reviewers were satisfied with this manuscript in its current form. While reviewer 1 did list a couple additional corrections that would be useful, one in particular should be attended to:

"Section 3.3.2 - These are great but the results are in the supplementary material. When I read this into the discussion section I was unclear what parameter set was being used for the following tables/figures. It should be made clear here or in the beginning of Section 4 that the results in the main manuscript are for the default scenario with the others in the supplementary material."

We have added a remark to this effect at the end of section 3.3.2.

Reviewer comments to Author:

Reviewer: 1

Comments to the Author(s)

Overall Review:

The authors have addressed most of my previous concerns. I remain curious about the initialization of the swarm which I explain below. However, the reason for the initialization has been addressed in this updated draft. Overall, the mild concern I had does not detract from the contribution of the paper and I believe it is suitable for publication.

Thank you for your helpful feedback.

I do have minor suggestions for this new draft that may improve the paper. I will list them below in the order they appear in the paper.

Initial condition concern from previous review and response:

Why are the robots so tightly packed initially, wont this essentially trap the robots in the middle initially forcing these results?

We would disagree that they are packed tightly, this would seem to imply they start in physical contact when in fact they have 7mm spacing (Methods 3.1).

In the context of an exploration or area coverage task, the scenario we have in mind is that they would be deposited in one location by a user, to then perform the job of spreading out.

To clarify, I did not mean to imply tightly packed was such that the robots could not move but that the initial pheromone deposition of the outer most agents would create a container for the inner robots (inner robots can only escape through 7mm gaps if none of the outside robots closed those gaps further. This may not have been a problem but hopefully clarifies my concern.

I do not think it is necessary to include another set of experiments but I still am interested if a different initial condition (long rectangle in stead of a square) would effect these results (as less agents would be in the center of the formation).

We have added a new remark about this in the Results and Discussion, lines 616-620.

Minor suggestions

Table 1: I am not a fan of "Here" as the reference for this paper, especially with the bottom ref cell being left blank. Maybe reformat the table with a top section for this paper's entries split with a double horizontal line for comparison to other papers.

This is a nit-picky suggestion. This table is a great addition to the paper as is!

We have tried to rearrange as suggested.

Line 279-280, the math here seems out of place and unnecessary. I think you could say in a rectangular arena half the area of the original with same aspect ratio (or something to that regard).

We have added your form of words, and we thought it was also helpful to include the actual dimensions of the arena.

Section 3.3.2 - These are great but the results are in the supplementary material. When I read this into the discussion section I was unclear what parameter set was being used for the following tables/figures. It should be made clear here or in the beginning of Section 4 that the results in the main manuscript are for the default scenario with the others in the supplementary material.

We have added a remark to this effect at the end of section 3.3.2.

Table 3/4 - The formatting of this table should be changed to make it visually cleaner and easier to read

We will work with the copy editors to improve readability of the tables.

Reviewer: 2

Comments to the Author(s)

The manuscript is much improved since its first version and I believe it is now ready for publication. Thanks very much for taking the time to include the suggestions of both reviewers. I think it makes for a stronger case for your hypothesis.

Thank you for your helpful suggestions for improvements.